# Vitamin D Pathway Genetic Variation and Type 1 Diabetes: A Case–Control Association Study

**DOI:** 10.3390/genes11080897

**Published:** 2020-08-05

**Authors:** Joana T. Almeida, Dircea Rodrigues, Joana Guimarães, Manuel C. Lemos

**Affiliations:** 1CICS-UBI, Health Sciences Research Centre, University of Beira Interior, 6200-506 Covilhã, Portugal; joanatralmeida@gmail.com (J.T.A.); joanaguimaraes.endoc@gmail.com (J.G.); 2C4-UBI, Cloud Computing Competence Centre, University of Beira Interior, 6200-501 Covilhã, Portugal; 3Serviço de Endocrinologia, Diabetes e Metabolismo, Centro Hospitalar Universitário de Coimbra, 3000-075 Coimbra, Portugal; dircearodrigues@chuc.min-saude.pt; 4Serviço de Endocrinologia, Centro Hospitalar do Baixo Vouga, 3810-193 Aveiro, Portugal

**Keywords:** type 1 diabetes, Vitamin D, autoimmune, genetics, single nucleotide polymorphism, SNP

## Abstract

Vitamin D has immunomodulatory effects, and its deficiency has been implicated in the autoimmune process of type 1 diabetes. Serum vitamin D levels are influenced by variants in genes involved in the synthesis, transport, hydroxylation and degradation of vitamin D. The aim of this study was to assess if single nucleotide polymorphisms (SNPs) at the *DHCR7* (rs12785878), *GC* (rs2282679), *CYP2R1* (rs2060793) and *CYP24A1* (rs6013897) loci are associated with type 1 diabetes in the Portuguese population. Genotype and allele frequencies were determined in 350 cases of type 1 diabetes and in 490 controls. The frequency of each SNP alone was not significantly different between patients and controls. However, the combined analysis of the four SNPs showed that minor alleles of these variants clustered more frequently in patients. The proportion of individuals with three or more minor alleles was significantly higher in patients than in controls (56.3% vs. 48.5; odds ratio (OR) 1.37; 95% confidence interval (CI) 1.04–1.81; *p*-value 0.027). These results suggest a cumulative effect of SNPs at the *DHCR7*, *GC*, *CYP2R1* and *CYP24A1* loci on the susceptibility to type 1 diabetes, due to the roles of these genes in the vitamin D metabolic pathway.

## 1. Introduction

Vitamin D regulates a variety of biological processes including the innate and adaptive immune systems [1]. Type 1 diabetes is characterized by an autoimmune destruction of the pancreatic islet beta cells, which results in a lack of insulin production [2]. Epidemiologic, pre-clinical and clinical studies have suggested that vitamin D deficiency is associated with islet autoimmunity and type 1 diabetes [3]. Observational studies have shown that vitamin D levels are frequently lower in patients with type 1 diabetes than in healthy controls [4]. In non-obese diabetic (NOD) mice, which are animal models for type 1 diabetes, vitamin D deficiency in early life leads to higher incidence and earlier onset of diabetes [5]. Conversely, early administration of vitamin D to NOD mice prevents the immune cell infiltration, pancreatic islet beta cell destruction and development of diabetes [3]. Likewise, in humans, vitamin D supplementation in infancy is associated with a reduction in the risk of type 1 diabetes [6,7]. Altogether, these findings suggest that impaired vitamin D metabolism may have a role in the pathogenesis of type 1 diabetes.

Serum levels of vitamin D depend on dietary intake and sunlight exposure. However, part of the variability of vitamin D levels is under genetic control and is explained by genetic variation of several components of the vitamin D pathway [8]. Genome-wide association studies (GWAS) have associated the variability in 25-hydroxyvitamin D levels with single nucleotide polymorphisms (SNPs) located in or near genes that are involved in the synthesis (7-dehydrocholesterol reductase, *DHCR7*), transport (group-specific component/vitamin D binding protein, *GC*), hydroxylation (cytochrome P450 subfamily IIR polypeptide, *CYP2R1*) and degradation (cytochrome P450 family 24 subfamily A polypeptide 1, *CYP24A1*) of vitamin D [9,10]. These GWAS demonstrated a strong association of the *DHCR7* rs12785878, *GC* rs2282679, *CYP2R1* rs2060793 and *CYP24A1* rs6013897 SNPs with serum 25-hydroxyvitamin D levels in individuals of European ancestry [9,10].

The aim of this study was to evaluate whether the *DHCR7* rs12785878, *GC* rs2282679, *CYP2R1* rs2060793 and *CYP24A1* rs6013897 SNPs are associated with the susceptibility to type 1 diabetes in the Portuguese population.

## 2. Materials and Methods

### 2.1. Subjects

The study was designed as a retrospective case–control association study. Cases comprised 350 patients with type 1 diabetes (184 males and 166 females; mean age at time of study ± standard deviation (SD) = 29.0 ± 11.1 years), recruited at diabetes outpatient clinics serving the central region of mainland Portugal. Mean age at diagnosis was 17.2 ± 9.4 years. Diagnosis of type 1 diabetes was based on classical clinical presentation, low or undetectable levels of serum C-peptide and presence of one or more pancreatic autoantibodies [2]. The control group comprised 490 unrelated healthy blood donors (246 males and 244 females, mean age ± SD = 32.2 ± 11.2 years), recruited at blood donation centres in the same region. All subjects were Caucasian Portuguese. The study was approved by the Ethics Committee of the Faculty of Health Sciences, University of Beira Interior (Ref: CE-FCS-2011-003 and CE-FCS-2013-017), and written informed consent was obtained from all subjects or their legal guardians.

### 2.2. Genetic Studies

The *DHCR7* rs12785878, *GC* rs2282679, *CYP2R1* rs2060793 and *CYP24A1* rs6013897 SNPs were analysed in this study because of their previous association with serum levels of vitamin D in individuals of European descent [9,10]. Minor allele frequencies (MAFs) for non-Finnish Europeans reported in population databases [11] were 0.278 (rs12785878 allele G), 0.288 (rs2282679 allele G), 0.403 (rs2060793 allele A) and 0.197 (rs6013897 allele A), respectively. Venous blood samples were collected from each subject, and genomic deoxyribonucleic acid (DNA) was extracted from peripheral blood leucocytes using previously described methods [12]. Genotyping of the rs12785878, rs2060793 and rs6013897 SNPs was performed by polymerase chain reaction-restriction fragment length polymorphism (PCR-RFLP) using previously described methods and custom-designed primers [13]. For each of these SNPs, 100 ng of genomic DNA was used in 15 μL reactions containing 0.25 μM of each primer, 1 U of Taq DNA polymerase and respective buffer (DreamTaq Green DNA polymerase, Thermo Fisher Scientific, Waltham, MA, USA) and 200 μM of each deoxynucleotide (dNTP). The PCR cycle conditions consisted of initial denaturation at 95 °C for 5 min, which was followed by 34 cycles of 95 °C for 30 s, 58 °C for 30 s and 72 °C for 30 s, with a final extension step of 72 °C for 10 min. Amplified fragments were then digested with the appropriate restriction enzyme (New England Biolabs, Beverly, MA, USA or Thermo Fisher Scientific, Waltham, MA, USA), according to the manufacturer’s instructions, and visualized after electrophoresis on a 2% or 3% agarose gel. The identification of the rs12785878, rs2060793 and rs6013897 SNPs was based on the different fragments produced after digestion with the TaqI, SchI and TscAI restriction enzymes, respectively [13]. Genotyping of the *GC* rs2282679 SNP was performed by a TaqMan SNP genotyping assay with a commercially available TaqMan probe (Assay ID: C-26407519-10; Thermo Fisher Scientific, Waltham, MA, USA), according to the manufacturer’s instructions. Genotyping failure rates were 0.4%, 0.1%, 0.1% and 1.2% for rs12785878, rs2282679, rs2060793 and rs6013897, respectively. The genotyping methods were validated by sequencing representative samples for each genotype (GenomeLab TM GeXP, Genetic Analysis System; Beckman Coulter, Fullerton, CA, USA). All genetic studies were carried out at the CICS-UBI, Health Sciences Research Centre, University of Beira Interior.

### 2.3. Statistical Analysis

Allele and genotype frequencies in the patient and control groups were determined by direct counting. Frequencies were compared by logistic regression analysis to obtain odds ratios (ORs), 95% confidence intervals (CIs) and p-values using the SNPassoc function (version 1.9–2) [14] in R software 3.4.4, adjusted for sex and age, to account for any effect of these variables on allele and genotype frequencies. The best model of inheritance for each SNP (dominant, recessive, codominant, overdominant or additive) was selected using Akaike information criterion. A Bonferroni correction for multiple comparisons was used to correct statistical significance, which was set at *p* < 0.0125 (*p* < 0.05, divided by the number of analysed SNPs). Testing for Hardy–Weinberg equilibrium was carried out by comparing the observed and allele-based expected genotype frequencies using a chi-squared goodness-of-fit test. Power analysis, using the software Power and Sample Size Calculations (version 3.1.6) [15], estimated that for MAFs between 20% and 40%, the sample size was sufficient to detect ORs between 1.49 and 1.56, for type 1 diabetes, under an additive model of inheritance, with a power of 0.8 and a type 1 error probability of 0.05. To determine the cumulative effect of minor (risk) alleles, each individual was classified as having zero, one or two minor alleles for each of the four SNPs. Individuals with missing data for one of the genotypes due to assay failure were excluded from this analysis. The total number of minor alleles per individual (ranging from a minimum of zero to a maximum of eight) was then compared between cases and controls using a two-tailed chi-squared test.

## 3. Results

The *DHCR7*, *GC*, *CYP2R1* and *CYP24A1* genotype and allele frequencies observed in patients and controls are presented in Table 1. No deviation from the Hardy–Weinberg equilibrium was observed, which would otherwise suggest selection bias, population stratification or genotyping errors. Near significant overrepresentations of the *DHCR7* and *CYP2R1* minor alleles were observed in patients (Table 1) but did not reach statistical significance after applying the Bonferroni correction for multiple comparisons (*p* > 0.0125).

The combined analysis of SNPs showed that type 1 diabetes patients had a higher number of minor alleles (Table 2). The proportion of individuals with at least three minor alleles was significantly higher in patients than in controls (56.3% vs. 48.5; OR 1.37; 95% CI 1.04–1.81; *p*-value 0.027).

## 4. Discussion

This case–control study did not detect a significant association of each individual SNP with type 1 diabetes but revealed a synergistic effect of multiple risk alleles. Individuals with type 1 diabetes were 1.4-fold more likely to have a combination of three or more minor alleles (OR 1.37; 95% CI 1.04–1.81). These results suggest that each SNP alone has a small undetectable effect, but the combination of several SNPs may result in an additive effect, contributing to a decrease of vitamin D levels and to an increase of risk for type 1 diabetes.

Most genetic studies of vitamin D and type 1 diabetes have focused on the vitamin D receptor (VDR) [16,17]. Few studies have analysed SNPs in the *DHCR7*, *GC*, *CYP2R1* and *CYP24A1* genes that have been associated with circulating vitamin D levels [9,10]. Cooper et al. [18] studied variants in *DHCR7*, *GC*, *CYP2R1* and *CYP24A1* in British patients with type 1 diabetes and found associations only with *DHCR7* and *CYP2R1*. In contrast, a study in the Danish population found no associations of these four loci with type 1 diabetes [19]. Another study in the North American population found an association of *DHCR7* (but not *GC*, *CYP2R1* or *CYP24A1*) with the development of islet autoimmunity [20]. A recent study in the Korean population showed an association between *CYP2R1* variants and type 1 diabetes [21]. These differences between studies suggest that geographical differences in diet, sun exposure and genetic backgrounds may compensate or aggravate the susceptibility conferred by variants in these genes [8].

Our study showed a higher frequency of the *DHCR7* rs12785878 and *CYP2R1* rs2060793 minor alleles in type 1 diabetes, resembling the results of Cooper et al. [18]. However, after applying the Bonferroni correction for multiple comparisons, our results were no longer statistically significant. Nevertheless, the combined number of *DHCR7*, *GC*, *CYP2R1* and *CYP24A1* minor alleles that each individual harboured was associated with the risk of type 1 diabetes in our population. This gene/allele dosage effect on the risk of type 1 diabetes has not been reported before and can be explained by a joint effect of these loci on vitamin D levels. The possibility of a gene/allele dosage effect is supported by two previous GWAS [9,10], which demonstrated that individuals who carried the highest number of risk alleles at these loci were more frequently affected by vitamin D deficiency.

The exact mechanisms through which these four SNPs, alone or in combination, influence circulating levels of vitamin D is presently unknown. These genetic variants are located in non-coding regions, and, so far, there are no functional studies to clarify if they exert a direct effect on gene expression or if they are in linkage disequilibrium with other functional variants. Further studies are needed to determine the functional consequences of these SNPs. In addition, the associations found in our study may be population-specific and not replicated in other populations with different environmental exposures and genetic profiles. Therefore, it would be of interest to replicate our study in other European and non-European populations.

A limitation of our study is the lack of serum vitamin D levels, as these are not routinely measured in the normal clinical management of patients with type 1 diabetes. However, measurements of serum vitamin D also have limitations, which include inter-laboratory differences, seasonal fluctuations, fasting status and time lapse between the development of the disease and the measurements. This means that single measurements of vitamin D levels do not necessarily reflect the levels to which individuals were exposed to before the onset of the disease. Therefore, the study of vitamin-D-related genes as surrogates for vitamin D status may provide a more stable indicator of vitamin D levels over the life course than measurement of serum vitamin D at one point in time.

In conclusion, our data suggest a synergistic effect of multiple alleles at the *DHCR7*, *GC*, *CYP2R1* and *CYP24A1* loci on the susceptibility to type 1 diabetes, due to the role of these genes in the vitamin D pathway and their effect on serum vitamin D levels. These observations may contribute to a better understanding of the role of vitamin D in the pathogenesis of type 1 diabetes.

## Figures and Tables

**Table 1 genes-11-00897-t001:** Distribution of *DHCR7*, *GC*, *CYP2R1* and *CYP24A1* genotypes and alleles in type 1 diabetes and controls.

Polymorphisms		Patients, *n* (%)	Controls, *n* (%)	OR (95% CI)	*p* Value	Adjusted OR (95% CI) ^†^	Adjusted *p* Value ^†^
**DHCR7 (rs12785878)**							
Genotypes	TT	114 (38.8)	189 (38.8)				
	TG	171 (48.9)	221 (45.4)				
	GG	65 (18.6)	77 (15.8)	1.31 (0.98–1.75) ^‡^	0.063	1.37 (1.01–1.85) ^‡^	0.042
Alleles	T	399 (57.0)	599 (61.5)				
	G	301 (43.0)	375 (38.5)	1.20 (0.99–1.46) ^§^	0.068	1.20 (0.98–1.47) ^§^	0.083
**GC (rs2282679)**							
Genotypes	TT	175 (50.0)	231 (47.2)				
	TG	155 (44.3)	220 (45.0)				
	GG	20 (5.7)	38 (7.8)	0.72 (0.41–1.26) ^¶^	0.242	0.70 (0.39–1.28) ^¶^	0.264
Alleles	T	505 (72.1)	682 (69.7)				
	G	195 (27.9)	296 (30.3)	0.88 (0.70–1.10) ^§^	0.264	0.83 (0.66–1.05) ^§^	0.123
**CYP2R1 (rs2060793)**							
Genotypes	GG	119 (34.0)	182 (37.2)				
	GA	169 (48.3)	249 (50.9)				
	AA	62 (17.7)	58 (11.9)	1.60 (1.09–2.36) ^¶^	0.018	1.45 (0.96–2.19) ^¶^	0.079
Alleles	G	407 (58.1)	613 (62.7)				
	A	293 (41.9)	365 (37.3)	1.22 (1.00–1.50) ^§^	0.055	1.19 (0.96–1.47) ^§^	0.114
**CYP24A1 (rs6013897)**							
Genotypes	TT	197 (56.6)	291 (60.4)				
	TA	129 (37.1)	172 (35.7)				
	AA	22 (6.3)	19 (3.9)	1.64 (0.88–3.09) ^¶^	0.121	1.65 (0.85–3.19) ^¶^	0.142
Alleles	T	523 (75.1)	754 (78.2)				
	A	173 (24.9)	210 (21.8)	1.19 (0.66–1.06) ^§^	0.139	1.19 (0.93–1.52) ^§^	0.165

*n*, number; OR, odds ratio; CI, confidence interval. ^†^ adjusted for sex and age; ^‡^ dominant model; ^§^ log-additive model; ^¶^ recessive model.

**Table 2 genes-11-00897-t002:** Distribution of the number of minor (risk) alleles in type 1 diabetes and controls.

Number of Minor (Risk) Alleles	Patients, *n* (%)	Controls, *n* (%)	OR (95% CI)	*p* Value
0	15 (4.3)	13 (2.7)		
1	43 (12.4)	87 (18.2)		
2	94 (27.0)	146 (30.5)		
3	93 (26.7)	120 (25.1)		
4	74 (21.3)	83 (17.4)		
5	20 (5.7)	23 (4.8)		
6	9 (2.6)	6 (1.3)		
7	0 (0.0)	0 (0.0)		
8	0 (0.0)	0 (0.0)		
2 or less	152 (43.7)	246 (51.5)	0.73 (0.55–0.97)	0.027
3 or more	196 (56.3)	232 (48.5)	1.37 (1.04–1.81)	0.027

*n*, number; OR, odds ratio; CI, confidence interval.

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
