# Peer review of "Vitamin D Pathway Genetic Variation and Type 1 Diabetes: A Case–Control Association Study"

_genes, 2020, doi:10.3390/genes11080897_

Round 1

Reviewer 1 Report

Vitamin D deficiency is commonly found in T1D patients and vitamin D is believed to exert immunomodulatory and anti-inflammatory effects to halt autoimmunity, hence addressing impaired metabolism of vitamin D and study its association with the disease proves important. Polymorphisms in genes that are critical for vitamin D synthesis and degradation were shown to module risks of the disease and biological significance of those polymorphisms warrants more studies to dissect its impact on disease pathogenesis. With this respect given study is timely to provide new knowledge on gene polymorphisms and associations with disease
Manuscript describes near significant association of specific single nucleotide polymorphisms (SNPs) in particular DHCR7 and CYP2R1 minor alleles in patients while combined analysis of SNPs including those in GC, CYP24A1 genes (besides those two mentioned above) shows enrichment of at least three minor alleles in patients when compared to controls in Portuguese population. These SNPs were selected based on previous literature showing a strong association of these SNPs with serum 25-hydroxyvitamin D levels in individuals of European ancestry. There is a major limitation in study that lacks knowledge on vitamin D levels in serum. Manuscript does not provide data from serum levels of vitamin D from individuals which is critical to know if given SNPs correlate and or have negative impact on vitamin D levels. Also, not clear from the methods how MAF (minor alleles) are calculated from genotypes, it is not clear if these are functional conserved SNPs, are minor alleles most likely to be risk alleles in this case ? What is biological relevance of SNPs found in none-coding region ?
Authors should clarify why mean age of T1D patients (at time of diagnosis) differ almost 2 fold from unrelated blood donors from same region. What is inclusion criteria for unrelated blood donors ? What are patient characteristics with respect to glycemic control, C-peptide levels and autoantibodies ? These are critical parameters to define and evaluate T1D pathogenesis, progression and clinical manifestation. 
Also will be of importance to at least discuss what is known on impaired vitamin D metabolism and its role on T1D with regard to malfunctioning of pancreatic beta cells, pancreatic beta cell islet destruction and immune cell infiltration.
To summarize, study provides initial indications of SNPs (involved in vitamin D metabolism) minor allele association with T1D but it also confuses reader on control group selection criteria, missing information on patients and lack of vitamin D level assessment, which makes it difficult to draw conclusions

Reviewer 2 Report

Reviewer comments and suggestions

The present study by Almeida et al. assessed the association among single nucleotide polymorphisms (SNPs) at the DHCR7 (rs12785878), GC (rs2282679), CYP2R1 (rs2060793) and CYP24A1 (rs6013897) loci with type 1 diabetes in the Portuguese population. The study was a retrospective case-control consisted of 350 cases of type 1 diabetes and in 490 controls. The study result discusses the combined analysis of the four SNPs revealed minor alleles of these variants clustered more frequently in patients. In conclusion, they reported that a synergistic effect of SNPs at the DHCR7, GC, CYP2R1 and CYP24A1 loci on the susceptibility to type 1 diabetes.

There are few comments to be responded before finalizing any decision on the MS. 

  1. In the subject section, line number 49, please specify the main place where the study was done.
  2. Line 50-51, for diagnosis, please specify the international criteria for this
  3. No information of sample collections, please mention
  4. Line 63-64, need to specify the method in a comprehensive way
  5. In table1 I saw most of the adjusted OR was not found to be significant. How does the author describe and conclude it
  6. Table 2 what was the differences between alleles 2 and 2 or less.
  7. Line 99 is there was a specific reason for this.
  8. Line 118-119, please specify the reason for the Danish population association with your study
  9. Line 130-131, The author need to describe it more
  10. Line 135-136 it seems a grammatical mistake, please correct the sentence
  11. Line 138, please explain the other similar study about the lack of serum vitamin D levels (not in limitation section you can add in the discussion part)
  12. Line 139, “However, measurements of serum vitamin D also have limitations and do not necessarily reflect the vitamin D status at the time of the development of the disease. could you elaborate it more

Reviewer 3 Report

Paper from Almeida et al. shows that the presence of multiple minor alleles, known as associated to a reduced Vit.D serum level, might have a synergistic moderate effect on type 1 diabetes risk, whereas each SNP alone has a small undetectable effect.

Statistical approach was adequate, result well delineated and discussion was appropriate and well calibrate to take home message that the study of SNPs influence on a multifactorial disease might be effective only if multiple and interactive gene variant are evaluated together at the same time.

Is a pity that Vitamin D dosages were unavailable as stratification of Vit.D serum levels on the bases of the presence of 3 or more minor alleles might have to add a good piece of evidence on the role of Vit.D genetically regulated metabolic pathway in T1DM susceptibility

Neverthless,  the manuscript seems to be suitable for publication in the present form 

Round 2

Reviewer 1 Report

Manuscript shows substantial improvement and adds more clarity on missing and confusing parts of the manuscript. It is unfortunate to have clinical data missing due to restricted access to patients' info. However, having discussed limitations, relevance of observed minor allele enrichment in patients and providing future perspectives in the manuscript will certainly help the reader to appreciate findings of the study. Hence, I support publication of the revised version of the manuscript